# Bone Metastases and Health in Prostate Cancer: From Pathophysiology to Clinical Implications

**DOI:** 10.3390/cancers15051518

**Published:** 2023-02-28

**Authors:** Cinzia Baldessari, Stefania Pipitone, Eleonora Molinaro, Krisida Cerma, Martina Fanelli, Cecilia Nasso, Marco Oltrecolli, Marta Pirola, Elisa D’Agostino, Giuseppe Pugliese, Sara Cerri, Maria Giuseppa Vitale, Bruno Madeo, Massimo Dominici, Roberto Sabbatini

**Affiliations:** 1Department of Oncology and Hematology, Azienda Ospedaliero Universitaria of Modena, 41125 Modena, Italy; 2Oncology, AUSL of Modena Area Sud, Sassuolo-Vignola-Pavullo, 41121 Modena, Italy; 3Oncology Unit 1, Department of Oncology, Veneto Institute of Oncology IOV—IRCCS, 35128 Padova, Italy; 4Department of Oncology, Azienda Ospedaliero Universitaria S. M. della Misericordia, 33100 Udine, Italy; 5Medical Oncology, Ospedale Santa Corona, 17027 Pietra Ligure, Italy; 6Department of Oncology and Hematology, Univerity of Modena and Reggio Emilia, 41125 Modena, Italy; 7Unit of Endocrinology, Department of Medical Specialities, Azienda Ospedaliero Universitaria of Modena, 41125 Modena, Italy

**Keywords:** bone metastasis, bone health, prostate cancer, bone health specialist, bone-targeted therapies

## Abstract

**Simple Summary:**

Bone metastases and health are crucial issues in prostate cancer patient management. The aim of our review was to describe the biology of bone metastases and their clinical implications in prostate cancer patients, and current therapeutic strategies. In addition, “bone health” should be evaluated and specific treatments considered. In this way, we aimed to attract attention to the risk of both “bad” bone health and bone metastasis, and also to the available care options for these patients.

**Abstract:**

Clinically relevant bone metastases are a major cause of morbidity and mortality for prostate cancer patients. Distinct phenotypes are described: osteoblastic, the more common osteolytic and mixed. A molecular classification has been also proposed. Bone metastases start with the tropism of cancer cells to the bone through different multi-step tumor–host interactions, as described by the “metastatic cascade” model. Understanding these mechanisms, although far from being fully elucidated, could offer several potential targets for prevention and therapy. Moreover, the prognosis of patients is markedly influenced by skeletal-related events. They can be correlated not only with bone metastases, but also with “bad” bone health. There is a close correlation between osteoporosis—a skeletal disorder with decreased bone mass and qualitative alterations—and prostate cancer, in particular when treated with androgen deprivation therapy, a milestone in its treatment. Systemic treatments for prostate cancer, especially with the newest options, have improved the survival and quality of life of patients with respect to skeletal-related events; however, all patients should be evaluated for “bone health” and osteoporotic risk, both in the presence and in the absence of bone metastases. Treatment with bone-targeted therapies should be evaluated even in the absence of bone metastases, as described in special guidelines and according to a multidisciplinary evaluation.

## 1. Introduction

Prostate cancer (PC) is the second most common cancer in men worldwide and more than half of PC occurs in men over the age of 70 years [1]. The propensity of PC cells to seed in the skeleton and then to progress into clinically relevant metastatic tumors is widely studied, and is a major cause of morbidity and mortality in PC patients [2]. Bone metastases most frequently affect the axial skeleton and often cause skeletal complications known as skeletal-related events (SREs), such as: pathological fracture, radiotherapy (RT), surgery, spinal cord compression (SCC) and hypercalcemia [3]. Despite the osteosclerotic nature of bone metastases, SREs in PC are still very common, reducing quality of life and worsening survival [3].

Bone metastases start with the tropism of cancer cells to the bone through specific migratory and invasive processes [4]. The complex molecular pathogenetic mechanism of bone metastases offers several potential targets for prevention and therapy [4].

Although the mechanisms underlying bone metastases are far from being fully elucidated, several translational models of PC bone metastases have been studied, including the application of molecular profiling techniques, animal model systems and engineered cell lines: all of these models could help to improve our treatment capacity.

Nowadays, several therapeutic options are available for PC patients. The milestone was androgen-deprivation therapy. Other possibilities now include chemotherapeutic agents, new-generation hormone therapies, radium 223 and, more recently, radioligand therapies. However, for these patients, special attention should be also placed on the management of bone health and the prevention of treatment-induced bone loss [3]. Bone-targeted agents, bisphosphonates and denosumab are active in bone metastases [1]; however, these drugs should still be evaluated even in the absence of bone metastases and under multidisciplinary evaluation, according to dedicated guidelines.

Our review aimed to attract attention to both the biological and clinical implications of bone metastases and to the risk of “bad” bone health in PC patients. 

## 2. Bone Metastases in Prostate Cancer

PC cells show a preference for tropism to the bone. An autopsy study revealed that approximately 90.1% of men who had died with metastases of PC were diagnosed with bone metastases [5]. In PC patients with bone metastases, the 5-year survival rate was 33% [6]. In cases of spinal metastases of PC, the median overall survival (OS) appears to be 24 months with an estimated 1-year OS of 73% [7]. The extent of skeletal metastatic involvement correlates with survival in patients with advanced PC. The “bone scan index” allows us to quantify the extent of tumor skeletal involvement. Patients with low, intermediate and extensive skeletal involvement had a median overall survival of 18.3, 15.8, and 8.1 months, respectively, in a study of 191 patients with androgen-independent PC [8]. 

Distinct phenotypes of bone metastases have been described in patients with PC: osteolytic, osteoblastic and mixed. The existence of mixed lesions suggests that the processes that regulate tumor-associated osteolysis and bone formation may occur together in bone metastases and are not mutually exclusive. Furthermore, the relative activity of these two coexisting processes defines the bone metastases’ phenotype. Osteolytic metastases, defined as a “punched-out” area of severe bone loss, are a consequence of tumor-induced activation of bone-matrix resorption. Resorption of mineralized bone matrix is the natural function of the osteoclast, a multinucleated cell of hematopoietic origin residing in the bone, in cooperation with multiple other actors and with several stimuli (as reported below). Osteoblastic metastases, characterized by bone forming, are prevalent in advanced PC patients and induced by cancer cell interactions with osteoblasts and their progenitors through several interactions [9]. PC cells also demonstrate osteomimicry by responding to growth factor stimulation [10]. This would suggest that bone-forming tumors may also occur through differentiation of the cancer cells towards an osteoblastic bone-forming phenotype, which is a phenomenon that has been observed in the bone metastatic PC cell line, C42b [11]. A category of cancer and bone interactions likely to contribute to the metastatic tumor phenotype are those driven by sex steroid hormones. Prostate and breast cancers, both sex steroid-sensitive diseases, show a predilection to form bone metastases. In addition, it has been shown that hormone-sensitive PC cells can respond to sex steroid deprivation by activating de novo synthesis [12], which implies that bone cells interacting with metastatic cancer may be stimulated by androgen produced locally by tumor cells. 

Osteoblatic metastases are more common in PC, representing 68% of all bone metastases [13]. Despite this, the osteolytic factor parathyroid hormone-related protein (PTHrP) is also highly expressed in PC. A proposed explanation is that PTHrP can also stimulate bone formation by activating the *ETAR* with NH2-terminal fragments of PTHrP, which share strong sequence homology with *ET-1* [14]. 

The prognosis of patients is markedly influenced by SREs, such as pathological fractures, hypercalcemia and pain, which occur in 49% of osteoblastic metastases [13]. To predict the risk of SREs, bone resorption markers may be useful, such as N-telopeptide of type I collagen (NTX) and bone alkaline phosphatase (BALP), which are associated with higher rates of death and SREs in PC bone metastases [15,16]. Further studies would be useful to stratify the risk of SREs in different types of bone metastases.

The field exploring potential biomarkers of bone metastases deserves special attention, and researchers have investigated new strategies and approaches with different biomarkers.

Yu and colleagues retrospectively analyzed data from 150 PC patients and found that patients with bone metastases had significantly elevated serum levels of carcinoembyonic antigen 125 (CA125), total prostate-specific antigen (T-PSA), free PSA (F-PSA), cytokeratin-19 fragment (CYFRA 21-1) and pro-gastrin-releasing peptide (ProGRP). The ROC curves indicated that T-PSA, F-PSA and ProGRP could effectively aid in discriminating between patients with bone metastases and those without. The area under the curves for the combination of these parameters was 0.941 with 90% sensitivity and gave better results than with each biomarker alone or with two biomarkers combined [17]. Instead, Aufderklamm and colleagues investigated the utility of serum c-terminal telopeptide of type I collagen (1CTP) and n-terminal propeptide of type I procollagen (P1NP) in the diagnosis of bone metastases and in the prognosis of patients. These peptides are markers of bone formation which are increased in PC patients and bone metastases. They analyzed serum samples of 186 patients with prostatic hyperplasia or PC, with or without metastases. Increased levels of 1CTP were found in PC patients compared with others, while no significant difference was shown for P1NP levels. Instead, both markers were altered in metastatic patients compared with non-metastatic ones. Cancer prognosis was significantly worse in metastatic PC patients with higher 1CTP concentration [18]. Moreover, to improve the capability of detecting for the risk of bone metastases, Windrichova and colleagues compared the performance of 16 biomarkers and suggested a mathematical model, the Bone Risk Score (BRS), by combining three of the biomarkers. They compared serum biomarkers levels in patients with different primary tumors, using scintigraphy to detect those with bone metastases (56 patients) or those without (75 patients). The best performance was obtained with the BRS combining P1NP, growth differentiation factor-15 (GDF15) and osteonectin [19]. In addition, Ku et al. carried out comprehensive expression profiling of tissue samples of bone metastasis from different types of cancer, revealing their proteome landscape and four significant proteins with the potential capability to differentiate tumor primaries [20]. Further studies are required to confirm these findings with a larger number of patients, and the clinical relevance of these markers.

## 3. Biology of Bone Metastases in Prostate Cancer

Two historical hypotheses to explain mechanisms of tumor dissemination are the “seed-and-soil” hypothesis proposed by Paget in 1889 [21] and the “mechanical entrapment theory” postulated by Ewing in 1928 [22]. After about a century, both of these hypotheses have been integrated and collected, according to the ‘metastatic cascade’ model [23], with different multi-step tumor–host interactions [24]. Its major steps are summarized in the following paragraphs and schematized in Figure 1 and Table 1.

Table 1, Step 1A describes tumor cells escaping from the primary tumor and preparing the metastatic niche by epithelial-to-mesenchymal transition (EMT) and exosome release. Step 1B describes invasion of the surrounding tissue, intravasation and survival into circulation (with platelet coat formation). Step 2 in Figure 1 and Table 1 describes implantation into the soil, in this case in the bone marrow. Tumor cells undergo arrest, extravasation and invasion, with settlement in the new tissue. In the bone marrow niche, as shown in the Figure 1 (step 2), tumor cells form relationships with multiple resident and attracted cells and several molecules (see text and Table 1). Step 3 in Figure 1 and Table 1 is the representation of dormancy. It is a particular phase of balance between tumor cells and normal cells; players and factors involved are multiple. Step 4 is growth, in which the results of multiple interactions may differ and the two extreme possibilities range from the predomination of the ‘osteoclastic vicious cycle’ (Step 4A) or the ‘osteoblastic vicious cycle’ (Step 4B), see Figure 1.

### 3.1. Prepare and Reach the Soil: The First, General Mechanisms

A.Escape from primary tumor and prepare the metastatic niche

Some tumor cells at the primary site undergo epithelial-to-mesenchymal transition (EMT) and release exosomes, small vesicles involved in cell-to-cell communication and expression of integrins capable of conditioning their target to create the pre-metastatic niche [25]. The suitable pre-metastatic niche must evolve to allow tumor cell engraftment (metastatic niche) and proliferation (micro- and macrometastatic transition) [26]. Fibroblasts, through fibronectin, attract hematopoietic cells from the bone marrow expressing vascular endothelial growth factor receptor-1 (VEGFR-1), and establish a metastasis-supporting microenvironment [27]. The primary tumor also releases VEGF-A, TGF-β and TNF-α, which, via expression of S100, mediate the migration of myeloid cells into the metastatic niche [24]. Matrix metalloproteinase (MMP) MMP-9 [28] and lysyl oxidase (LOX) [29] are also important for metastatic niche’s formation. 

B–C.Invasion of surrounding tissue and intravasation

MMP-1, -2, -7, -9 and -14 are involved in tumor angiogenesis [30] and MMP-14 may remodel the extracellular matrix (ECM) to facilitate cancer cell migration and invasion [31]. Tumor-associated macrophages (TAM) [32,33,34] and PHD2 expressed in the tumor vasculature are important in these steps [35]. Cancer cells can access the bloodstream directly through compromised tumor-associated blood vessels; or with active intravasation [36], dysregulation of angiogenesis [37]; or through inflammatory signaling [38,39]. 

D–E.Survival in circulation and “attraction” to new locations

Cancer cells may be isolated from the blood stream as circulating tumor cells (CTCs), either as single cells or as clusters [40]. To reach their goal, CTCs have to survive into the circulation and act in cooperation with platelets, which adhere to their surface, increasing their metastatic potential, preventing their recognition by the immune system [41] and decreasing their shear stress [42]. Stromal cells secrete several chemokines such as CXCL12 and RANKL [43,44] to “attract” cancer cells to the bone marrow.

### 3.2. Implant in the Soil: Prostate Cancer Cell Homing in the Bone Marrow

F.Arrest

The mechanical entrapment of CTCs in capillaries preludes their arrest in the tissue [24]. Platelets help in the initial adherence [45] and facilitate the initial interaction of cancer cells through E-selectin, which is expressed in the endothelium [46] and in the primary cancer [47]. Platelets have been implicated in the specific development of bone metastases through the release of lysophosphatidic acid and the production of IL-6 and IL-8, stimulating osteoclast activity [48]. Integrins, CD44 and MUC1 are then important, for a more stable interaction of CTCs with the endothelium [49]. PC cells bind preferentially to the bone marrow endothelium [50]. Jung and colleagues, using an in vivo murine model of human PC cell metastasis, noted that growth arrest specific-6 (GAS-6) levels were significantly greater in the forelimb versus hindlimb bone marrow, and spinal lesions or lesions in the bones of the hindlimb were more frequent than those of the forelimb [51]. GAS-6 is a ligand for the tyrosine kinase receptor Axl, and its role in prostate cancer is controversial but the therapeutic manipulation of its levels may prove useful for treatment of metastatic disease. 

G.Extravasation

Once bound to the endothelium, cancer cells begin opening the endothelial junctions in response to multiple factors—including TGF-β and VEGF [52]—traverse the basement membrane and enter into the stroma. The endothelial cells of the bone perivascular niche modulate cell trafficking. 

H.Settlement

Once they arrive in the bone marrow, cancer cells require phenotypical changes to stabilize and survive. They acquire the capability to respond to physical and chemical microenvironmental stimuli and adhere to the special niches previously prepared. The CXCL12/CXCR4 axis facilitates the bone invasion processes by inducing MMP-9 and downregulating the expression of tissue inhibitors of MMP-2 in PC cells [53,54,55]. Moreover, MMPs lead to cancer cell colonization in the bone marrow, through integrin αvβ3 and integrin αvβ5, which interact with osteopontin and integrin-binding sialoprotein (IBSP), respectively [56,57,58]. Galectin-3/Thomsen–Friedenreich antigen is one other adhesion molecule important for the interaction of PC cells with bone marrow endothelium [59]. The protein CCL5, a member of the chemokine superfamily produced by cells in the bone microenvironment, with its receptor CCR5, increases PC cell migration to the bone via androgen receptor signaling [60]. Other molecules are also involved [61,62,63]. Furthermore, the transcription factors Twist-1 and lysophosphatidic acid (LPA) are reported to be important for bone invasion due to their expression of two microRNAs, miR-10b and miR-21, respectively. Knocking out these two microRNAs inhibits bone marrow invasion in in vivo experiments [64,65]. 

### 3.3. Dormancy: The Prelude of Detectable Bone Metastases

In the bone niche, cancer cells have to engage in several interactions with stromal and resident cells, resulting in different outcomes: a ”silent balance” called “dormancy” or an “activation state” that lead to an “osteoclastic” or an ”osteoblast vicious cycle” (Figure 1, detail). In 1998, Luzzi et al. provided one of the first descriptions of these interactions, underlining the multistep nature of metastatic inefficiency and two critical points: failure of solitary cells to initiate growth and failure of early micrometastases to continue growth [66]. Among others, GAS-6, bone morphogenetic protein 7 (BMP7) and transforming growth factor beta 2 (TGF-β2) have been associated with dormancy. GAS-6 acts by binding to receptors Axl, Sky and Mer [67]. Shiozawa et al., demonstrated that the activation of Axl by GAS-6 on PC cells in a bone marrow niche environment plays a critical role as a molecular switch to establish dormancy of PC cells [67]. BMP7 and TGF-β2 act via inhibitions of the ERK signaling pathway [68]. Cancer-associated fibroblasts (CAFs) and the immune system are important for cancer cell survival in bone marrow [69,70] and the immune system is a key player in tumor cell dormancy [71]. NK cells, via production of interferon γ (INFγ) and TRAIL/FASL-induced apoptosis, collaborate to maintain balance [72]. 

### 3.4. Growth: The “Clinical Phase” of Bone Metastasis 

In triggering cancer cell reactivation after dormancy, the important players are: endothelial cells from neovasculature that produce TGF-β1 and periostin [73]; adipocytes via FABP4 [74]; macrophages with cathepsin K [75]; and protein in the microenvironment, such as type 1 collagen [76] and fibronectin [77]. Furthermore, immature myeloid cells (iMCs) in the presence of tumor cells differentiate into myeloid-derived suppressor cells (MDSCs) and TAM [78]. Dendritic cells (DCs) play a critical link between innate and adaptive immunity and the inhibition of a special population such as plasmacytoid DCs is associated with a greater Th1 response, INFγ production and restoration of CD8+ T cell function against cancer cells [79]. In the bone, neutrophils (TAN) enhance bone resorption [80]. In this scenario, cancer cells mimic bone cells in a manner called osteomimicry. They have been shown to have an osteoblast-like phenotype, owing to their expression of cathepsin K, osteonectin, cadherin-11, connexin-43 and RUNX2 [81,82]. However, cancer cells may also acquire osteoclast properties due to fusion with macrophages, or induce multinucleated giant cells due to fusion with osteoclast precursors [83]. Moreover, cancer cells may shift their behavior more towards growth by their expression of VCAM1 and the release of signals related to the NFkB pathway [84]. Further research will allow us to specify even more detail about the role and relationships between resident and circulating cells, both in cancer and non-cancer cells, and factors used to communicate. Some important growth factors are reported below. 

IGF is the most abundant growth factor stored in bone, and metastatic PC cells are positive for IGF type I receptor (IGF-IR). Their interaction increases proliferation and cancer cell survival through AKT and NF-κB signaling [85]. 

TGF-β, the second most abundant growth factor stored, promotes the production of osteolytic factors that induce RANKL expression and inhibit osteoprotegerin expression in BMSCs and osteoblasts. The latter promote osteoclastic bone destruction; progression of bone metastases; and also secrete several proteins that positively regulate tumor growth, including IL-6, SPARC and periostin. SPARC induces cancer migration and homing through the αVβ5 integrin, whereas periostin and IL-6 promote prostate tumor survival [86]. Moreover, high levels of extracellular calcium facilitate bone metastases of PC via the CaSR and the Akt signaling pathway [87]. PC cells, on their side, produce several cytokines and growth factors, including IL-6, BMP, TGFR, VEGFR and Wnt. These factors activate osteoblasts, which secrete RANKL. RANKL binds to receptor activator of NF-κB (RANK) expressed on osteoclasts, resulting in osteoclast activation. Osteoclasts reabsorb bone and release growth factors supporting tumor growth, such as TGF-β. Osteoprotegerin (OPG), an inhibitor of RANKL, is consequently overwhelmed by TGF-β and unable to oppose RANKL production, continuing the vicious cycle of bone metastases [88]. PTHrP was shown to potently stimulate osteoclastogenesis by increasing the production of RANKL by osteoblasts. However, PTHrP also facilitates osteoblastic alterations [89]. 

Recently, the role of the Wnt pathway in the progression of prostate bone disease has been investigated. The Wnt gene family is a big family of cysteine-rich glycoproteins. In this context, Wnt induces osteoblastic activity through upregulation of OPG expression and downregulation of RANKL, which together increases the osteoblastic phenotype of bone metastases [90]. Hall et al. demonstrated that elevated glycoprotein Dickkopf-1 (DKK-1) is an early event in prostate cancer, and a decline is a later event in advanced bone disease. The decline of DKK-1 levels in bone metastases is interlinked with the osteoblastic activity of Wnt and supports a model in which DKK-1 is a molecular switch that transitions the phenotype of PC bone lesions from osteolytic to osteoblastic. DKK-1 has proved to be oncogenic and an inhibitor of Wnt signaling and, thus, of bone formation (osteoinduction) [91].

Sclerostin is another protein secreted by osteocytes and was recently shown to both upregulate the expression of RANKL by osteocyte-like cells and promote osteoclastogenesis [92].

Last but not least, the role of androgen receptor (AR) should be underlined. During the castration-resistant phase of prostate cancer, AR is reactivated through several mechanisms, including AR amplification and mutation, as well as activation of ARs through other signaling pathways. Bone metastases usually occur in the castration-resistant phase and androgen receptor variants (AR-Vs), active even in the absence of ligand-binding domain (including AR-V1, AR-V7, and AR-V567es), are highly expressed in bone metastases of patients with castration-resistant PC (CRPC). Cellular-myelocytomatosis viral oncogene (c-Myc) has a positive role in regulating ARs and AR-Vs in prostate cancer as reported by Bai et al., in cell models and in a patient-derived xenograft model [93]. Importantly, their study highlights the role of c-Myc in CRPC and suggests the utility of its target as an adjuvant to AR-directed therapy. They demonstrated that the inhibition of c-Myc sensitizes enzalutamide-resistant cells to growth inhibition by enzalutamide, one of the second-generation anti-androgen therapies used for PC treatment [93]. The close relationship between c-Myc and ARs was recently underlined also by Qiu and colleagues. They demonstrated that c-Myc overexpression significantly diminishes the AR transcriptional program and contributes to PC initiation and progression [94]. 

#### 3.4.1. Osteolytic Lesions: The ‘Osteoclastic Vicious Cycle’

Reactivated cancer cells express VCAM-1 with the recruitment of osteoclastic precursors and the release of several factors, such as PTHrP. This process increase the expression of RANKL, responsible of the formation of new osteoclasts [95]. Myeloid bone marrow cells and lymphocytes produce cytokines that stimulate osteoclast activity [95]. Osteoblast activity is inhibited by cancer cells through the release of soluble proteins, such as DKK-1 [95]. Bone tissue may contribute to osteolysis by the production of growth factors, such as TGF-β and IGFs I and II [95].

#### 3.4.2. Sclerotic Lesions: The “Osteoblast Vicious Cycle”

In their paper, Logothetis et al. reported the role of BMP-2, Wnt, adrenomedullin, FGF9, endothelin-1 and OPG in osteoblast activity in PC with bone metastasis [9]. 

#### 3.4.3. Mixed Lesions

The division between the two previously mentioned types of bone lesions is not well defined. In a single patient, and even in a single lesion, they may co-exist and produce mixed lesions. 

## 4. Molecular Subtypes of Prostate Cancer Bone Metastases: Beyond “Classical” Characteristics of Bone Metastases

The important role of ARs in PC has already been emphasized, and CRPC bone metastases can be divided into two subgroups, according to AR activity: high and low AR activity subgroups. These two groups of bone metastases have different immune cell profiles [96,97]. Moreover, following analysis of genome-wide expression (GWAS) within PC bone metastases from patients who were untreated or who underwent androgen-deprivation therapy (ADT), Thysell and colleagues identified three distinct molecular subtypes within bone lesions: Met A, B and C. The subtypes have different gene expression, morphology and clinical features (Table 2). MetA is the most frequent, has a high expression of androgen receptor-regulated genes, including Prostate-Specific Antigen (PSA) and seems to be of luminal cell origin. MetB shows poor prognosis after ADT, has a dedifferentiated luminal phenotype and some characteristics similar to neuroendocrine tumors. It exhibits profiles related to DNA damage and cell cycle activity, androgen-stimulated gene expression is generally low and cell proliferation is high. MetC shows high transcription activity involved in stroma–epithelial cell interactions and inflammation. This latter group is the least common and most poorly defined [98]. Using PSA and Ki67 analysis, the same group was able to differentiate MetA-like from MetB-like tumors, with different prognoses (Table 2) [98]. 

Recently, the same group verified the clinical relevance of the MetA-C subtype classification, and in particular its usefulness to identify MetB patients in need of complementary therapy [99]. They retrospectively analysed a total of 103 metastasis samples from 67 clinically different PC patients and from the sequencing data of 573 other metastasis samples previously published. Their results confirmed that MetA was the most common subtype and had a high androgen response, while MetB was associated with poor prognosis; was enriched in CRPC and in liver metastases; was characterized by high cell cycle activity and DNA repair; and demonstrated specific gene alterations. MetC was characterized by epithelial-to-mesenchymal transition and inflammation, and showed diverse biology, organ tropism and prognoses [99]. 

Moreover, researchers examined whether bone metastatic subtypes and prognosis after ADT could be predicted by immunohistochemical analysis of epithelial and stromal cell markers in primary tumor biopsies made at diagnosis [100]. They analysed samples from primary tumors and metastases from 98 PC patients and found that International Society of Urological Pathology (ISUP) grade was not associated with outcome or metastasis subtypes, whereas high expression on tumor epithelial cells of Ki67 in combination with low PSA expression and a low fraction of AR positive stroma cells, correlated with poor prognosis after ADT and with developing of MetB subtypes. The opposite pattern predicted the development of the MetA subtype with better ADT response. Thanks to these results, a subtype-specific metastasis treatment could be initiated at diagnosis, for example, complementary therapies for patients with a primary tumor with high proliferation and/or low PSA expression. It is noteworthy that the analysis was restricted to microscopic evaluation and that the correlation coefficient for individual factors measured in primary tumors and paired metastases samples were all relatively low. The quantification of “hot spot” regions in primary tumors could have higher correlation; there is also the consideration of other aspects, such as the bone microenvironment. Furthermore, it remains to be explored how this conclusion applies to different patients, such as those diagnosed at an earlier disease stage [100]. 

Further studies are needed to clarify whether patients with different metastasis types would benefit from different therapies or new subtype-specific treatments. 

## 5. Systemic Treatments in Prostate Cancer and Skeletal Related Events

ADT with surgical bilateral orchiectomy, administration of non-steroidal anti-androgens or analogs of luteinizing hormone releasing hormone (LHRH) are the possible therapeutic options administered in PC in different settings [101]. The goal of ADT is to reduce testosterone by up to 95% and to lower estrogens, but it also results in increased bone resorption in order to alter the balance between osteoblastic and osteoclastic cells, and a rapid decline of bone mineral density (BMD). The duration of ADT is proportional to the risk of osteoporotic fracture [102]. In contrast, treatment with peripheral anti-androgens does not cause bone adverse events [103]. ADT is also related to modification of body composition: loss of muscle mass (sarcopenia) and an increase in fat mass [102,104,105,106,107]. 

Management of bone health and prevention of cancer treatment-induced bone loss (CTIBL) in an important part of the treatment of PC patients undergoing hormonal treatment [1], and prevention of CTIBL is covered by already-cited ESMO guidelines [3].

When PC becomes resistant to androgen deprivation (CRPC), the disease is more aggressive and often metastatic. Optimal management of PC patients with bone metastases requires a multidisciplinary team composed of a medical oncologist, radiotherapist, orthopedic specialist, interventional radiologist, nuclear medicine physician and bone specialist. 

In CRPC patients, LHRH therapy is combined with second-generation hormonal therapies, such as abiraterone [108], enzalutamide [109], apalutamide [110] or darolutamide [111], or with chemotherapeutic drugs such as docetaxel [112] or cabazitaxel [113]. Metabolic radiotherapy can also be used in patients with metastatic prostate cancer and symptomatic bone metastases [114]. 

There are no data about the role of taxane-based chemotherapy in the control of SRE, but it induces myelosuppression [112,113], and, in animal models, the administration of drugs with medullar toxicity were responsible for persistent loss of trabecular components of bone and increased bone resorption [107,115]. Moreover, the use of over-physiological glucocorticoids in patients undergoing taxane chemotherapy could adversely affect bone health. Glucocorticoids inhibit osteoblastic differentiation and increase osteoclastic survival, promoting bone resorption [116]. 

Oral abiraterone acetate plus prednisone compared with placebo and prednisone improved OS (15.8 mo vs. 11.2 mo; *p* < 0.0001), delayed time to first SRE (9.9 mo vs. 4.9 mo, *p* = 0.0001), median time to occurrence of first SRE (25 vs. 20.3 mo, *p* = 0.0001), enhanced pain relief and improved quality of life (QoL) in metastatic CRPC (mCRPC) previously treated with docetaxel in the COU-AA-301 trial [108,117]. In the COU-AA-302 trial, where treatment with abiraterone acetate plus prednisone compared with placebo and prednisone was evaluated in mCRPC patients who had not previously received chemotherapy, the time to first SRE was not among the endpoints; however, the drug improved radiographic progression-free survival (PFS) and significantly delayed clinical decline [118]. The STAMPEDE trial analyzed the role of abiraterone acetate plus prednisolone and ADT compared with ADT alone in patients with locally advanced or metastatic PC. After 3 years of treatment, the combination arms showed an elevated survival rate (83% vs. 76%, HR 0.63; *p* < 0.001) and a reduced risk of SRE (12% vs. 22%, HR 0.46, *p* < 0.001) [119].

Enzalutamide, an oral non-steroidal antiandrogen evaluated in the AFFIRM trial, improved OS in comparison with placebo (18.4 mo vs. 13.6 mo) and showed a reduction in the risk of a first SRE (16.7 mo vs. 13.3 mo) in metastatic prostate patients after docetaxel-based chemotherapy [120]. The PREVAIL trial showed an improved time to first SRE (32% vs. 37%, HR, 0.72; *p* < 0.001) in metastatic chemotherapy-naïve patients treated with enzalutamide compared with placebo (median 31.1 mo vs. 31.3 mo) [109]. In men with nonmetastatic CRPC with rapidly rising PSA levels at high risk for metastases, enzalutamide (in comparison with placebo) significantly lowered the risk of metastases and death in the PROSPER trial [121].

Apalutamide in men with nonmetastatic CRPC at high risk for the development of metastases had significantly improved metastasis-free survival and time to symptomatic progression compared with placebo in the SPARTAN trial [110]. 

Darolutamide is an antagonist of the androgen receptor. Its role has been evaluated in the ARAMIS trial in men with non-metastatic CRPC. It delayed the time to first appearance of a symptomatic skeletal event versus placebo (16 events vs. 18 events, HR 0.43, *p*: 0.01) [111]. 

Radium-223-dichloride is a bone-targeting agent approved for patients with symptomatic bone metastases from PC without visceral disease. Radium-223-dichloride binds with high-affinity hydroxyapatite in sites with elevated bone turnover, and has a local cytotoxic effect via double-strand DNA breaks [114]. In the ALSYMPCA trial, patients with symptomatic bone metastases of PC after docetaxel or not suitable for docetaxel were treated with six cycles of intravenous radium-223-dichloride. Compared with placebo, the experimental group showed better OS (14.9 months vs. 11.3 months; HR, 0.70 *p* < 0.001) and a longer time to first SRE [114,122]. Patients receiving antiresorptive treatments in addition to radium-223-dichloride showed a delayed time to first symptomatic SRE and a prolonged symptomatic SRE-free survival time (HR 0.69, *p* < 0,0001) [123]. However, the results of the ERA 223 trial should be emphasized: radium-223-dichloride plus abitaterone acetate and prednisone did not reduce the risk of SKE or improve survival in mCRPC, but they did increase the risk of fracture. After a median of 21.2 months, at least 1 SKE was reported in 49% of patients in the experimental group vs. 47% of controls, and the median symptomatic skeletal event-free survival was 22.3 vs. 26.0 months. Moreover, 29% of patients followed for safety in the experimental group experienced bone fracture, especially osteoporotic fractures, compared with 11% of controls [124]. This evidence led the US Food and Drug and the European Medicines Agency to revise the prescribing recommendations for radium-223-dichloride, but it should be stressed that patients in both treatments arms who used bone health agents had a reduced risk of fracture compared with non-users. 

The results of studies of the beta-emitting lutetium (Lu)-177-labeled prostate-specific membrane antigen (PSMA) radioligand therapy (RLT) for mCRPC were presented in 2021 with the phase 3 randomized trial VISION. 177Lu-PSMA-617 plus standard care compared with standard care alone significantly prolonged imaging-based PFS and OS (primary end points). The time to first SRE was also longer: 11.5 vs. 6.8 months with an HR of 0.50 (95% CI, 0.40–0.62) *p* < 0.001. The incidence of high-grade adverse events was also higher with 177Lu-PSMA-617 than without, but quality of life was not adversely affected [125].

In mCRPC that progressed after Lu-177-PSMA treatment, measurable antitumor effects were seen also with alpha-emitting actinium (Ac)-225-PSMA-617 RLT [126]. 

For metastatic hormone-sensitive PC (mHSPC) several studies demonstrated that the addition of abiraterone (LATITUDE [127] and STAMPEDE [119]), apalutamide (TITAN [128]), enzalutamide (ARCHES [129] and ENZAMET [130]) or docetaxel (CHAARTED [131] and STAMPEDE [132]) to ADT improves OS, but also in this setting the attention to bone health and SRE should not be underestimated. 

Table 3 reports phase II/III trials on prostate cancer and outcomes in terms of OS and time to first SRE.

## 6. Bone Health in Prostate Cancer

### 6.1. Bone Loss 

Osteoporosis is a skeletal disorder characterized by decreased bone mass and qualitative alterations associated with increased fracture. Bone Mineral Density (BMD), evaluated by dual-energy x-ray absorptiometry (DXA), represents an accurate and precise measurement of bone mass. Fracture risk exponentially increases at a T score < −2.5 SD, which has been established by the WHO as the cut-off for densitometric diagnosis of osteoporosis [115]. There is a close correlation between osteoporosis and PC. From 3.9% to 37.8% of hormone-naïve PC patients show osteoporosis before the start of any oncological treatment, suggesting that PC could itself be a risk factor for loss of BMD, due to the promotion of bone resorption [133]. Osteoporosis in patients treated with LHRH analogs involving any site varies from 10% to 40% and worsens with age and ADT duration [133,134,135], reaching 80% of patients after 10 years of treatment [134]. The annual rate of bone loss in all men is between 0.5% and 1%. Bone loss during the first year of ADT in patients with metastatic disease is 2–8% in the lumbar spine [135,136] and 1.5–6.5% in the hip [136,137,138,139]. In subsequent years, the reduction in BMD continues, at approximately 1–4% each year [134]. After bilateral orchiectomy, BMD at the femoral neck diminishes by 2.4% after the first year and by 10% after two years [140]. At the end of ADT, BMD may increase in the lumbar spine, while remaining low in other sites [136,140,141], no increase is observed at the hip. Risk factors for bone loss are older age and lower body mass index (BMI) [135,139,142]. Bone loss is associated with an increased risk of incident fractures [137]. 

### 6.2. Fracture Risk

PC is not an independent risk factor for bone fractures [143]. After bilateral orchiectomy, BMD diminishes and the fracture rate is 38% at 5 years [139]. All patients treated with ADT should be evaluated for osteoporotic risk during treatment. The main risk factors are age ≥ 75 years, history of low-energy fracture after the age of 50 years, osteoporosis defined as a T-score ≤ −2.5 at one or both measurement sites (spine and femur), BMI < 19 kg/m^2^, at least three comorbidities (e.g., cardiovascular disease, depression, Parkinson’s disease, dementia) and current or past glucocorticoid therapy [144,145,146]. Several trials showed that treatment with an LHRH analog for longer than 6 months is associated with an increased fracture risk. In men older than 50 years old without PC, osteoporotic fracture risk was 13% versus 21–37% in patients with PC [107,116]. Patients who received LHRH treatment suffered bone fractures in 19.4% of cases from 1 to 5 years after diagnosis, versus 12.6% without LHRH analog therapy (*p* < 0.001) [102]. The relative risk (RR) of fracture in each bone was 1.21 (*p* < 0.001), 1.45 for vertebral fractures (*p* < 0.001) and 1.30 for hip fractures (*p* = 0.002) [106]. Fracture risk was associated with mortality risk [146]. The risk of fall was increased by loss of muscle mass secondary to decrease in testosterone levels [147]. The FRAX score can be used to estimate the absolute 10-year risk of osteoporotic hip fracture and major osteoporotic fracture (clinical spine, forearm, hip or shoulder fracture) in men older than 40 years old. This tool could drive physicians to start treatment for osteoporosis. In men on ADT, Adler et al. underline that DXA and FRAX identify ADT-treated men differently for treatment for osteoporosis [148]. The Italian Association of Clinical Endocrinologists (AME) position statement for the treatment of osteoporosis recommends considering for treatment all subjects with a BMD assessment T-score ≤ −2.5 SD with prior fragility fracture, regardless of BMD measurement, or with a DXA-based T-score between −2.5 and −1 SD and with an increased 10-year fracture risk evaluated with a fracture risk algorithm [149]. In contrast, Dawson-Hughes and colleagues in the practice guidelines published in the USA in 2008, suggested a cost-effective cut-off for the treatment of osteoporosis when the 10-year probability of hip fracture reached 3% [150]. 

### 6.3. Treatment for Bone Health in PC Patients

SRE-like pathological fractures, spinal compression, bone pain and increased levels of calcium are involved in around 40% of patients with metastatic PC, and influence QoL [151]. In every man before starting ADT and during antineoplastic treatment, for maintaining bone health, Body Mass Index (BMI), medical history (for example, diabetes, smoking history, alcohol abuse, use of medications such as steroids), history of fractures, dietary calcium intake, physical activity, vitamin D, and calcium and phosphorous levels are topics to investigate (Figure 2). The NCCN guidelines for PC version 2.2021 for screening and treatment of osteoporosis in patients on ADT refer to the National Osteoporosis Foundation guidelines recommending calcium and vitamin D3 supplementation and additional treatment (denosumab, zoledronic acid, alendronate) for men aged ≥50 years with low bone mass (T-score between −1.0 and −2.5) at the femoral neck, total hip or lumbar spine by DEXA and a 10-year probability of hip fracture ≥3% or a 10-year probability of a major osteoporosis-related fracture ≥20% (fracture risk assessed using FRAX algorithm) [152]. The European Academy of Andrology (EAA) clinical guideline on management of bone health published in 2018 as regards PC patients receiving ADT recommends starting antiresorptive treatment in patients with moderate-to-high fracture risk (with FRAX score) [153]. The already-cited position paper of AME for the treatment of osteoporosis suggests that men on ADT perform mild endurance exercise consistent with their overall clinical state; consume 1000–2000 mg daily calcium, possibly from their diet; receive vitamin D supplementation if they have low plasma levels; and alendronate or zoledronate if they have a high risk of fracture; denosumab is also recommended [149]. The same experts recommend against the use of selective estrogen receptor modulators (SERMs) for treating men with ADT, as these drugs are not registered for this indication [149]. 

Most trials evaluated the role of biphosphonates in preventing BMD decline, but they were not powered to evaluate fracture risk reduction [154,155]. On the other hand, denosumab, a human monoclonal antibody associated with RANKL inhibition that suppresses bone resorption caused by osteoclasts [156], is associated with increased BMD [157] and also demonstrated a significant reduction in the incidence of new vertebral fractures at 36 months in men on ADT and one additional risk factor for fracture (age > 70, T-score < 1.0 or history of osteoporotic fracture) [158]. In their study, Smith et al. found a BMD increase of 5.6% in the lumbar spine, 4.8% at the total hip and 3.9% in femoral neck (*p* < 0.001) and a reduced risk of incident vertebral fracture over 36 months with denosumab (1.5% vs. 3.6% in placebo arms; RR 0.38, 95% CI 0.17–0.78) [158].

For patients with SRE, treatment with antiresorptive drugs should be evaluated. Bisphosphonates and denosumab are active in bone metastases for their suppression of bone resorption and they improve bony tenderness and pain. Intravenous bisphosphonates showed a longer duration of action than the oral formulation [159]. Bisphosphonates cause osteoclast death during bone resorption [160]. In locally advanced or recurrent castration-sensitive PC, the upfront use of bisphosphonates did not show a survival benefit [132,161]. In patients with CRPC with bone metastases, treatment with intravenous zoledronic acid every 21 days showed a reduction in SRE [162] and improved BMD in lumbar spine (8.09%, 95% CI 5.89–10.29; *p* < 0.00001), in total hip (4.45%, 95% 0.84–8.06%, *p* = 0.02) and in femoral neck [159]. Denosumab 120 mg monthly showed superior results when compared with zoledronic acid (4 mg monthly) for prevention of SRE in men with bone metastatic CRPC (HR 0.82; CI 0.71–0.95; *p* = 0.0002) [163]. 

Antiresorptive drugs are well tolerated and adverse events, such as fever, myalgias and atypical femoral fractures, are rare. Jaw osteonecrosis (ONJ) is a possible adverse event when bisphosphonates and denosumab are administered for long time in patients with low oral hygiene, prior tooth surgery or who use a dental device. The risk is very low at the dosages used for osteoporosis, and slightly greater when used for bone metastases, but it remains infrequent and its management is mostly conservative [164]. In patients with risk factors for ONJ, preventive dental treatments are indicated before bone-target therapy and education of clinicians and patients about oral health before and during antiresorptive therapy may help reduce the incidence of ONJ and improve its outcomes [164]. Before every intravenous infusion, tests for serum creatinine clearance, calcium and phosphorus levels must be completed and adequate vitamin D supplementation should be ensured. If creatinine clearance level is between 30 and 60 ml/min then bisphosphonates can be administered with caution. 

The optimal duration of antiresorptive treatments is still unclear and must be evaluated for each patient, both to prevent bone loss and to treat bone metastases. In clinical trials evaluating the prevention of bone loss, bisphosphonates were administered for different durations [165,166,167,168,169,170,171,172,173]. In cancer patients with bone metastases continuing treatment beyond 2 years there may be some benefit, in addition to an individualized approach and a switching strategy after skeletal disease progression [174]. In patients with different cancers with bone metastases, the administration of i.v. zoledronic acid every 12 weeks is non-inferior to 4-week schedules, with a similar incidence of SRE ≥1 within 2 years of randomization [175]. Nevertheless, for each case, a multidisciplinary team evaluation is desirable. Additionally, for example, in patients with oligometastatic disease, low risk of SRE and good response to systemic treatment, antiresorptive drugs may have a limited duration of effect; therefore, in cases of multiple bone metastases, bisphosphonates or denosumab may instead be administered for longer if well tolerated. Palliative radiation therapy is indicated for patients with metastatic disease with bone pain. Steroid treatment can be administered in case of an initial flare in bone pain. Radiotherapy demonstrates rapid resolution of pain and an overall response rate (ORR) of 70–80%. It improves QoL and it is related to a low rate of adverse events [176]. 

## 7. Conclusions 

Several therapeutic options are available for PC patients, but bone metastases are still a relevant problem both for morbidity and mortality. Researchers on biological mechanisms for their formation and growth and on their molecular landscape could offer several potential targets for prevention and therapy. However, it is also important to raise awareness in the oncology and medical community of the maintenance of bone health before and during oncological treatments. Every man with PC should be evaluated for osteoporosis risk before starting ADT. Patients with an elevated risk of osteoporosis and “bad” bone health should be referred to a multidisciplinary panel that includes a bone health specialist. For these patients, treatment with bone-targeted therapies should be evaluated, even in the absence of bone metastases. In this way, we can try to ensure the best care for PC patients, with significant improvement both in quality of life and in overall survival.

Bone health plays a central role in PC. It may be influenced by cancer treatments and several other conditions and may be divided in two aspects: bone metastasis and osteoporosis. An appreciation of the biology of bone and bone metastases is important for understanding and choosing treatment strategies. ‘Bad’ bone health conditions can lead to major skeletal events. Their prompt recognition and treatment, with the help of a multidisciplinary team with a bone health specialist, can improve the quality of life and survival of patients.

## Figures and Tables

**Figure 1 cancers-15-01518-f001:**
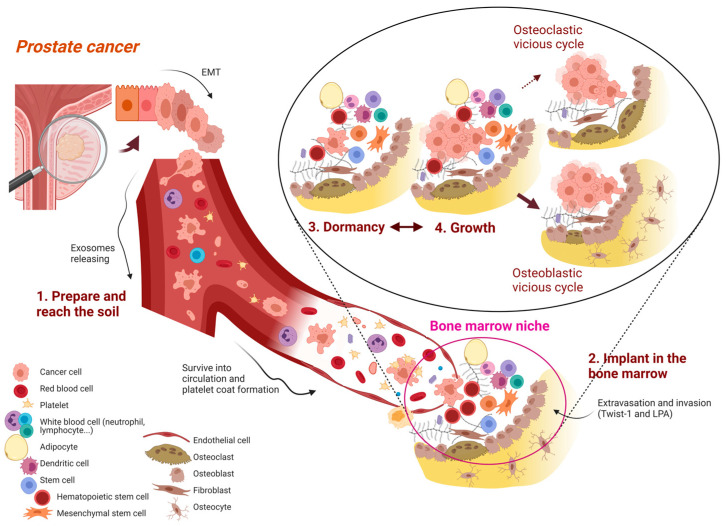
Steps of metastatic cascade.

**Figure 2 cancers-15-01518-f002:**
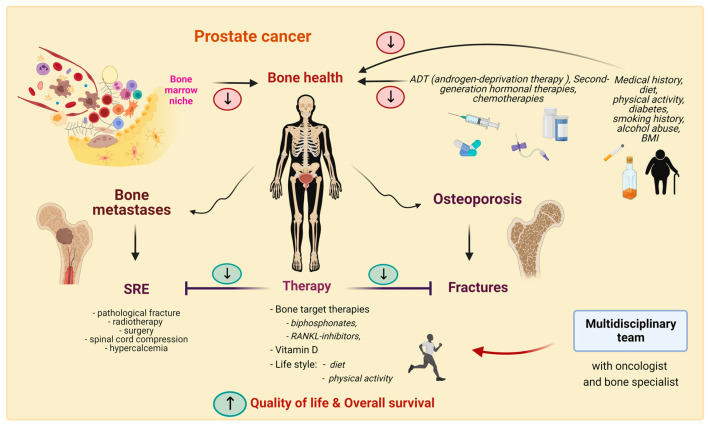
Possibilities and strategies for bone health in PC. The green ovals indicate interventions or factors that improve bone health, while the red ovals indicate factors that make it worse.

**Table 1 cancers-15-01518-t001:** The metastatic cascade’s major steps.

Process			Cells Other than Cancer Cells	Molecules	
				From Tumor	From Other Cells
**1**	**Prepare and Reach the Soil**				
	A	Escape from primary tumor and prepare metastatic niche	Fibroblasts; Hematopoietic stem cells	Exosomes with integrins; *VEGF-A*, *TGF-β* and *TNF-α*; *MMP-9*, *LOX*	*Fibronectin* *VEGFR-1*
	B	Invasion of surrounding tissue	TAM	*MMP-1,2,7,9,14*	*MMP*
	C	Intravasation	TAM; vasculature		*PHD2*
	D	Survival in circulation	Platelets		
	E	“Attraction” to new locations	Stromal cells	*CXCR4* *RANK*	*CXCL12* *RANKL*
**2**	**Implant into the Soil**				
	F	Arrest	Platelets; Endothelial cells		*Lysophosphatidic acid, IL-6, IL-8; E-selectin, integrins, CD44, MUC1*
	G	Extravasation	Endothelial cells	*TGF-β, VEGF*	*Adhesion molecules*
	H	Settlement	Stromal cells	*CXCR4, MMP-2, MMP-9, Integrin αvβ3, αvβ5* *CCR5*	*CXCL12* *Galectin-3/Thomsen-Fr Ag* *CCL5*
**3**	**Dormancy**		CAF, NK cells	*Osteomimicry*	*GAS6, BMP7, TGF-β2;* *INFγ, TRAIL-FASL*
**4**	**Growth**		Endothelial cells; Adipocites; Macrophages; MDSC and DC; TAN	*Osteomimicry with osteoblast-like phenotype or osteoclast properties;* *VCAM1, NFkB*	*TGF-β1; periostin; FABP4*; *Cathepsin K; Collagen t.1, fibronectin*
	I	Osteoclastic lesion	Pro-osteoclasts and osteoclasts; Myeloid cells and lymphocytes	*VCAM1, PTHrP, DKK-1*	*TGF-β, IGF-*1
	II	Slerotic lesions	Osteoblasts	*OPG, BMP-2, Wnt, adrenomedullin, FGF9, PDGF, ET-1*	*IL-6, MCP-1, VEGF, MIP-2*
	III	Mixed lesions			

**Table 2 cancers-15-01518-t002:** Molecular subtypes of prostate cancer bone metastases [98].

Subtypes	N of Cases	Cellular Differentiation	Gene Expression	Ki-67	PSA Level	Prognosis
MetA	71%	Moderate cellular atypia, glandular differentiation	KLK3, FOXA1, KRT18, CDH1	Low	High	Good
MetB	17%	Prominent cellular atypia, lack of glandular differentiation	FOXM1, CCNB1-2, CDC25B, CDK1, PLK1, PKMYT1, LMB1, KNSL1, NCL, KRT18 and others	High	Low	Poor
MetC	12%	Prominent cellular atypia, glandular differentiation detectable in some cases, relatively high stroma/epithelial ratio	ECM remodelling, regulation of EMT (Wnt, Notch, TGF-β, PDGF, immunological synapse formation, C/EBP, GSTP1	Low	Low	Poor

**Table 3 cancers-15-01518-t003:** Systemic treatments in prostate cancer, overall survival (OS) and time to the first skeletal-related event (SRE).

Author	Trial	Drug	Setting	N° of Patients	OS	*p*-Value	Time to First SRE *	*p*-Value
Tannock et al., 2004 [112]	TAX 327	Docetaxel (3 weekly and w) + prednisone vs. Mitoxantrone (m) + prednisone	mCRPC	1006 (335 vs. 334 vs. 337)	18.9 vs. 17.4 vs. 16.5	0.009 (3 w vs. m); 0.36 (w vs. m)	No data	-
Sweeney et al., 2015 [131]	CHAARTED	Docetaxel + ADT vs. ADT	mHSPC	790 (397 vs. 393)	57.6 mo vs. 44.0 mo	<0.001	No data	-
De Bono et al., 2010 [113]	TROPIC	Cabazitaxel + prednisone vs. Mitoxantrone (m) + prednisone	mCRPC	755 (378 vs. 377)	15.1 mo vs. 12.7 mo	<0.0001	No data (bone pain 5% vs. 5%)	-
Logothetis et al., 2012 [117]	COU-AA-301	Abiraterone + prednisone vs. placebo + prednisone	mCRPC	1195 (797 vs. 398)	15.8 mo vs. 11.2 mo	<0.0001	9.9 mo vs. 4.9 mo	0.0001
James et al., 2017 [119]	STAMPEDE	Abiraterone + prednisone + ADT vs. ADT	mHSPC and mCRPC	1917 (960 vs. 957)	83% vs 76% (3-year OS rate)	<0.001	12% of events vs. 22% of events	<0.001
Fizazi et al., 2017 [127]	LATITUDE	Abiraterone + prednisone + ADT vs. placebo + ADT	mHSPC	1199 (597 vs. 602)	not reached (NR) vs. 34.7 mo	<0.001	NR vs NR	0.009
Scher et al., 2012 [120]	AFFIRM	Enzalutamide vs. placebo	mCRPC	1199 (800 vs. 399)	18.4 mo vs. 13.6 mo	<0.001	16.7 mo vs. 13.3 mo	<0.001
Beer et al., 2014 [109]	PREVAIL	Enzalutamide vs. placebo	mCRPC	1717 (872 vs. 845)	32.4. mo vs. 30.2 mo	<0.001	32% events vs. 37% events	<0.001
Armstrong et al., 2019 [129]	ARCHES	Ezalutamide + ADT vs. placebo + ADT	mHSPC	1150 (574 vs. 576)	NR (HR 0.81)	0.3361	NR (HR 0.52)	0.0026
Davis et al., 2019 [130]	ENZAMET	Ezalutamide + standard care vs. standard care	mHSPC	1125 (563 vs. 562)	NR (at 36 mo: 80% vs. 72%)	0.002	No data	-
Chi et al., 2019 [128]	TITAN	Apalutamide + ADT vs. placebo + ADT	mHSPC	1052 (525 vs. 527)	NR (at 24 mo: 82.4% vs. 73.5%)	0.005	NR (HR 0.80)	-
Fizazi et al., 2019 [111]	ARAMIS	Darolutamide vs. placebo	non mCRPC	1509 (955 vs. 554)	NR vs. NR	0.045	16 events vs. 18 events	0.01
Parker et al., 2013 [114]	ALSYMPCA	Radium-223-dichloride vs. placebo	mCRPC	921 (614 vs. 307)	14.9 mo vs. 11.3 mo	<0.001	15.6 mo vs. 9.8 mo	<0.001
Smith et al., 2019 [124]	ERA 223	Radium-223-dichloride vs. placebo in addition to Abiraterone + prednisone	mCRPC and bone met	806 (401 vs. 405)	30.7 mo vs. 33.3 mo	0.128	22.3 mo vs. 26.0 mo	0.2636
Sartor et al., 2021 [125]	VISION	177Lu-PSMA-617 plus standard care vs. standard care	mCRPC	831 (551 vs. 280)	15.3 mo vs. 11.3 mo	<0.001	11.5 mo 6.8 mo	<0.001

* Or similar, e.g., median time to next symptomatic skeletal event, median symptomatic skeletal event-free survival.

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
