# Peer review of "Bone Metastases and Health in Prostate Cancer: From Pathophysiology to Clinical Implications"

_cancers, 2023, doi:10.3390/cancers15051518_

Round 1

Reviewer 1 Report

The manuscript by Baldessari et al., is a review of the current up-to-date state of knowledge about bone metastases arising from prostate cancer. The review is well-written and analyzes both the causes of bone metastases and the therapies and trials that are currently being used and in development. The bibliography is very useful and extensive.

My following comments are of minor character:

- Lines 17: Simple Summary is missing

- Line 37: Here and throughout, please use an abbreviation for “prostate cancer”. For instance, PCa or PC….

- Line 114: Where is Figure 1?

- Lines 410-411: I don't understand the meaning of this sentence

- Line 557: Where is Figure 2?

- Author contributions and Conflicts of interest are missing

Author Response

The manuscript by Baldessari et al., is a review of the current up-to-date state of knowledge about bone metastases arising from prostate cancer. The review is well-written and analyzes both the causes of bone metastases and the therapies and trials that are currently being used and in development. The bibliography is very useful and extensive.

My following comments are of minor character:

- Lines 17: Simple Summary is missing  Thank for your suggestion. We added Simple Summary.

- Line 37: Here and throughout, please use an abbreviation for “prostate cancer”. For instance, PCa or PC… Thank for your suggestion. We inserted the abbreviation.

- Line 114: Where is Figure 1? Thank for your suggestion. We added Figure 1 inside the text.  

- Lines 410-411: I don't understand the meaning of this sentence. Thanks. We re-wrote the sentence.

- Line 557: Where is Figure 2? Thank for your suggestion. We added Figure 2 inside the text.

- Author contributions and Conflicts of interest are missing. We added this part

Reviewer 2 Report

The paper by Baldessari et al. in an interesting overview about the correlation between pathophysiology and clinical implications of Bone metastases and health in prostate cancer.

The topic of the paper is worthy of investigation and well fits with the cope of the journal. It is an opinion of this reviewer that the paper can be published, and no specific comments/improvements seems to be needed. The only suggestion is related to last section of the paper, where authors are encouraged to improve the critical discussion about the issue covered and the clinical implications of their findings. Moreover, authors should consider the possibility to add some schemes/figures within the text to help readers in the flux of information.

Author Response

Thank for your suggestion. We added figures inside the text and modified the last section of the paper.

Reviewer 3 Report

This manuscript reviews prostate cancer-associated bone metastases, and the title indicates that the focus is the pathophysiology and clinical implications. The manuscript is not ready for a peer-review process because of a missing simple summary, as well as missing Figure 1 and Figure 2. The abstract is too vague for almost no meeting, finishing the sentence with the need for a significant improvement not only in quality o life but also in overall survival. The conclusion has little connection to the main review body, stating the importance of smoking history and alcohol abuse.

Major

·         Missing simple summary

·         Missing figure 1

·         Missing figure 2

·         Illogical abstract: The statement, “… essential to develop an improved treatment strategy. Even more, it is important to raise awareness …” These sentences mean that it is more important to raise awareness than improve a treatment strategy. The whole abstract hardly means anything because of vague descriptions.

·         Conclusion: the conclusion is detached from the main review body.

·         Introduction: Many sentences are not logically connected in the introduction. Lines 46-48 for the description of targeted therapies do not connect to lines 49-51 for the description of non-targeted therapy. Lines 54 to 58 are the justification of the title, but the review is not properly focused on this introductory description.

Minor

·         Lines 64 to 66: it is unclear why these numbers are described.

·         Too many abbreviations: they make the reading very difficult.

·         Link to osteoporosis: this is an important item because of osteolytic outcomes with osteoblastic metastases. The review is undigested, and no clear message is provided.

Author Response

This manuscript reviews prostate cancer-associated bone metastases, and the title indicates that the focus is the pathophysiology and clinical implications. The manuscript is not ready for a peer-review process because of a missing simple summary, as well as missing Figure 1 and Figure 2. The abstract is too vague for almost no meeting, finishing the sentence with the need for a significant improvement not only in quality o life but also in overall survival. The conclusion has little connection to the main review body, stating the importance of smoking history and alcohol abuse.

Major

  • Missing simple summary. We added this part. Sorry for the forgetfulness.
  • Missing figure 1. We added Figure 1 inside the text.
  • Missing figure 2. We added Figure 1 inside the text.
  • Illogical abstract: The statement, “… essential to develop an improved treatment strategy. Even more, it is important to raise awareness …” These sentences mean that it is more important to raise awareness than improve a treatment strategy. The whole abstract hardly means anything because of vague descriptions.

Thank for your suggestion. We re-wrote the whole abstract.             

  • Conclusion: the conclusion is detached from the main review body.

Thank for your suggestion. We revised this part of the text.              

  • Introduction: Many sentences are not logically connected in the introduction. Lines 46-48 for the description of targeted therapies do not connect to lines 49-51 for the description of non-targeted therapy. Lines 54 to 58 are the justification of the title, but the review is not properly focused on this introductory description.

Thank for your suggestion. We revised this part of the text.

Minor

  • Lines 64 to 66: it is unclear why these numbers are described. We want to underline the important impact of bone metastases.
  • Too many abbreviations: they make the reading very difficult. Sorry, we built an abbreviation index.
  • Link to osteoporosis: this is an important item because of osteolytic outcomes with osteoblastic metastases. The review is undigested, and no clear message is provided.

We hope that the changes made to the text clarify the message.

Reviewer 4 Report

Dear Editors,

The review by Baldessari and colleagues focuses on bone metastases and bone health in patients with prostate cancer. The Authors analyse, as a starting point, the biology of bone metastases and then highlight the therapeutic strategies useful for counteracting prostate cancer metastases and maintaining bone health.

This review is well-written and documented.

However, in my opinion, sections 3.4.1 and 3.4.2 (osteolytic lesions and osteosclerotic lesions) should be better explained.

There are also some minor points to clarify.

- Page 2, lines 96-98: could you please specify better the sentence?

- Figure 1: indicate the name of the main cells presented in the figure.

- I think the number 4 in Table 1 refers to "Growth", so it should be put next to "Growth"

- Figure 2: Specify what the green and red arrows mean.

Best regards,

P.M.

Author Response

However, in my opinion, sections 3.4.1 and 3.4.2 (osteolytic lesions and osteosclerotic lesions) should be better explained. Thanks for the suggestion. We modified this part.

There are also some minor points to clarify.

- Page 2, lines 96-98: could you please specify better the sentence? We modified the sentence.

- Figure 1: indicate the name of the main cells presented in the figure.  We modified the figure, thanks.

- I think the number 4 in Table 1 refers to "Growth", so it should be put next to "Growth" thank, we corrected the mistake.

- Figure 2: Specify what the green and red arrows mean. In the green arraws we indicated intervention that improve bone health, while in red arrows that make it worse. We specificated it in the test.

Reviewer 5 Report

This review was written well. However, I cannot find Figure 1 and 2 in manuscript file. Could you show us?

Author Response

Thank you for the comment. We added figures inside the text.

Reviewer 6 Report

Comments to the authors: manuscript "Cancers-2203527"

The authors submitted a well-structured and comprehensive overview. I have only two suggestions:

1.    The issue of subtypes of bone metastases in prostate cancer (MetA - C) should be updated considering some new study data, see Mol Oncol 2022,16:846-59 and Cancers 2022,14:5195. Table 2: reference should be included in the Table heading or in a footnote.

2.    The authors only very briefly addressed the issue of circulating bone markers (line 99-106). It would be useful and complete the review to include a brief separate chapter assessing critically the clinical relevance of bone markers based on corresponding reviews and considering new strategies/approaches like new biomarkers, technologies etc. (Clin Chim Acta 2020,510:437-41; Transl Androl Urol 2021;4000-8; Translat Cancer Res 2020,.2390-401; Technol Cancer Res Treat 2018;17:.1533033818807466).

Author Response

The authors submitted a well-structured and comprehensive overview. I have only two suggestions:

  1. The issue of subtypes of bone metastases in prostate cancer (MetA - C) should be updated considering some new study data, see Mol Oncol 2022,16:846-59 and Cancers 2022,14:5195. Thank you for your suggestion, it improves our review. We added and updated this part.

 Table 2: reference should be included in the Table heading or in a footnote. We added reference.

  1. The authors only very briefly addressed the issue of circulating bone markers (line 99-106). It would be useful and complete the review to include a brief separate chapter assessing critically the clinical relevance of bone markers based on corresponding reviews and considering new strategies/approaches like new biomarkers, technologies etc. (Clin Chim Acta 2020,510:437-41; Transl Androl Urol 2021;4000-8; Translat Cancer Res 2020,.2390-401; Technol Cancer Res Treat 2018;17:.1533033818807466). Thank you for your suggestion. We added this part

Reviewer 7 Report

The paper is interesting, however, few mistakes have to be corrected before it may be considered for publication:

1. Please provide a simple summary "Submissions without a simple summary will 22 be returned directly. Example could be found at https://www.mdpi.com/2076-2615/6/6/40/htm.".

2. Figure 1 is missing.

3. Idea to divide text in paragraphs 3.1 and 3.2 into subsections using A-H is very confusing for the reader. Please make the text more coherent - maybe combining it with the table before would make more sense?

4. Line 228 "Some other factors, role and relationships between resident and circulating cells, remain 228 incompletely understood.". Please discuss those other factors. 

5. Conclusions section should be written again. There should be no direct reference to the figures nor the figures themselves. You should not introduce unknown factors in this section. 

7. Figure 2 is missing and probably should be in another section than the conclusions. 

6. Please add the authors' contributions.

7. Please give a statement about the conflict of interest. 

8. Please add email for correspondence.

Author Response

The paper is interesting, however, few mistakes have to be corrected before it may be considered for publication:

  1. Please provide a simple summary "Submissions without a simple summary will 22 be returned directly. Example could be found at https://www.mdpi.com/2076-2615/6/6/40/htm.". Thank for your suggestion. We added Simple Summary.
  2. Figure 1 is missing. We added Figure 1 inside the text
  3. Idea to divide text in paragraphs 3.1 and 3.2 into subsections using A-H is very confusing for the reader. Please make the text more coherent - maybe combining it with the table before would make more sense? Thank for your suggestion. The using of letter A-H in subparagrapher resumes the division in the table explaining in more detail the process.
  4. Line 228 "Some other factors, role and relationships between resident and circulating cells, remain 228 incompletely understood.". Please discuss those other factors. We reformulate the sentence, some factors are discussed below, further researches will allow us to specify even more details about the role and relationships between resident and circulating cells both cancer and non-cancer cells and factors used to communicate.
  5. Conclusions section should be written again. There should be no direct reference to the figures nor the figures themselves. You should not introduce unknown factors in this section. We re-wrote this part. Figure and a sentence were added to their specific paragraph (‘treatment for bone health in prostate cancer patients’)
  6. Figure 2 is missing and probably should be in another section than the conclusions.  We added Figure 2 inside the text.
  7. Please add the authors' contributions. We added this part. Sorry for the forgetfulness.
  8. Please give a statement about the conflict of interest. We added this part. Sorry for the forgetfulness.
  9. Please add email for correspondence. We added this part. Sorry for the forgetfulness.

Round 2

Reviewer 7 Report

The authors have addressed my comments and now the paper may be considered for publication.